# Sparse estimation of mutual information landscapes quantifies information transmission through cellular biochemical reaction networks

Swarnavo Sarkar [1✉], Drew Tack[1] & David Ross [1✉]

Measuring information transmission from stimulus to response is useful for evaluating the signaling fidelity of biochemical reaction networks (BRNs) in cells. Quantification of information transmission can reveal the optimal input stimuli environment for a BRN and the rate at which the signaling fidelity decreases for non-optimal input probability distributions. Here we present sparse estimation of mutual information landscapes (SEMIL), a method to quantify information transmission through cellular BRNs using commonly available data for single-cell gene expression output, across a design space of possible input distributions. We validate SEMIL and use it to analyze several engineered cellular sensing systems to demonstrate the impact of reaction pathways and rate constants on mutual information landscapes.

---

[1] National Institute of Standards and Technology, Gaithersburg, MD 20899, USA. ✉email: swarnavo.sarkar@nist.gov; david.ross@nist.gov

All living systems need to sense and respond to environmental changes for survival. At the cellular level, biochemical reaction networks (BRNs) accomplish the necessary sensing and response functions by controlling the conditional expression of genes. Studies of natural BRNs highlight the importance of sensing to optimally adjust gene expression in naturally evolved systems. More recently, advances in synthetic biology have enabled the design of synthetic BRNs and the construction of hierarchical modular arrangements of BRNs[1]. These new capabilities to engineer BRNs offer enormous potential for living therapeutics[2] and other applications of programmed sensing[3]. However, reliable engineering of BRNs will require quantitative metrics for performance and robust methods to evaluate those metrics from experimental data.

BRNs allow cells to react to unpredictable environments by transmitting information about input stimuli into cellular output response[4]. Consequently, as a performance metric for engineered BRNs, one should ask, how much information do they transmit? Information theory provides the only general metric for this purpose: the mutual information between the input and the output, which is a logarithmic measure of the number of distinguishable output response levels[5–8]. Mutual information has emerged as an important aspect of biology as we search for quantitative principles to unify our understanding of signal transmission in gene expression[9], differentiation[10], cell death[11], and other biological processes[7,8,12,13].

Notably, information transmission depends both on the probability distribution of the input and the conditional probability distribution of the output for each possible input. The role of stochasticity in BRN outputs is mainly studied using the magnitude of noise[14,15], but the evaluation of mutual information would provide a universally applicable way to study the impact of a stochastic input environment on the state of a BRN[16]. With mutual information, different BRNs could be compared independent of their biological context using the same metric, and the same, comparable units (e.g., bits). However, conventional methods for computing mutual information cannot be directly applied to most commonly available data, which consist of output distributions measured at a relatively small number (~10) of fixed input values, because those methods either require data at a large number of densely spaced input values[17] or a parametric model of the output response[18].

Here, we present sparse estimation of mutual information landscapes (SEMIL), a method to determine the information transmission through BRNs using commonly available data such as flow cytometry or single-cell microscopy. SEMIL finds the best discrete and sparse approximation for each possible continuous input probability distribution[19], which enables estimation of information transmission using BRN output data for a small number of discrete input values (see Methods, Supplementary Methods 1, and Supplementary Fig. 1). SEMIL produces mutual information landscapes that quantify the performance of engineered BRNs across a design space, consisting of the input probability distributions within which the BRNs are targeted to function (Fig. 1). We validate the accuracy of SEMIL using simulated output from model BRNs, and demonstrate the utility of SEMIL using engineered BRNs in bacterial cells.

## Results

**Validation of SEMIL using model response functions**. To validate SEMIL, we first used simulated data for a model response function approximating the gene expression output of a BRN using a gamma distribution with parameters that depend on the value of the input (Fig. 2a and Supplementary Methods 2). We used this model to generate mock data to analyze using SEMIL and compared the estimated mutual information with the correct results obtained via numerical integration. The comparison demonstrates that SEMIL can provide accurate estimates of mutual information (typically within 0.05 bits) even with data from a set of only five discrete input values (Fig. 2b, c). The accuracy further improves (typically within 0.02 bits in the high information transmission region) if the set of input values is increased from five to ten. Larger error (~0.2 bits) only occurs when the input distribution cannot be well approximated by the set of inputs at which the output data is available, for example, when the geometric mean of the input distribution is comparable to the lowest discrete input value (near the left boundary of Fig. 2c).

We further evaluated SEMIL by comparing with an exact result for information transmission in the small noise limit: the cumulative distribution function of the optimal input distribution, at which information transmission is maximal, converges to the mean input–output response function with decreasing noise in the output[16]. Therefore, we considered three model input–output response functions with the same mean response function but decreasing magnitudes of noise (green lines in Fig. 2d–f and Supplementary Methods 3). For each of the three

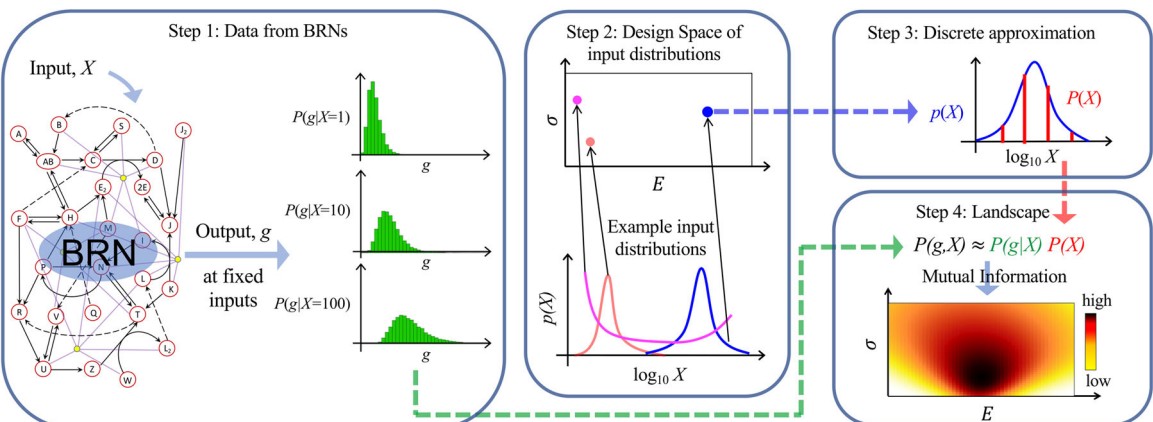

**Fig. 1 Estimating mutual information of BRNs with SEMIL.** Scheme for computing mutual information with SEMIL using output distributions for a sparse set of input values. Step 1: Data are acquired for the distribution of BRN output at a set of fixed input values. Step 2: A design space is specified to cover the range of input probability distributions within which the BRNs are targeted to function. Each point in the design space corresponds to a possible input probability distribution. Step 3: The best discrete approximation is identified for each input distribution in the design space. Step 4: The output distributions and the discrete approximation to the input distribution are used to calculate the mutual information landscape.

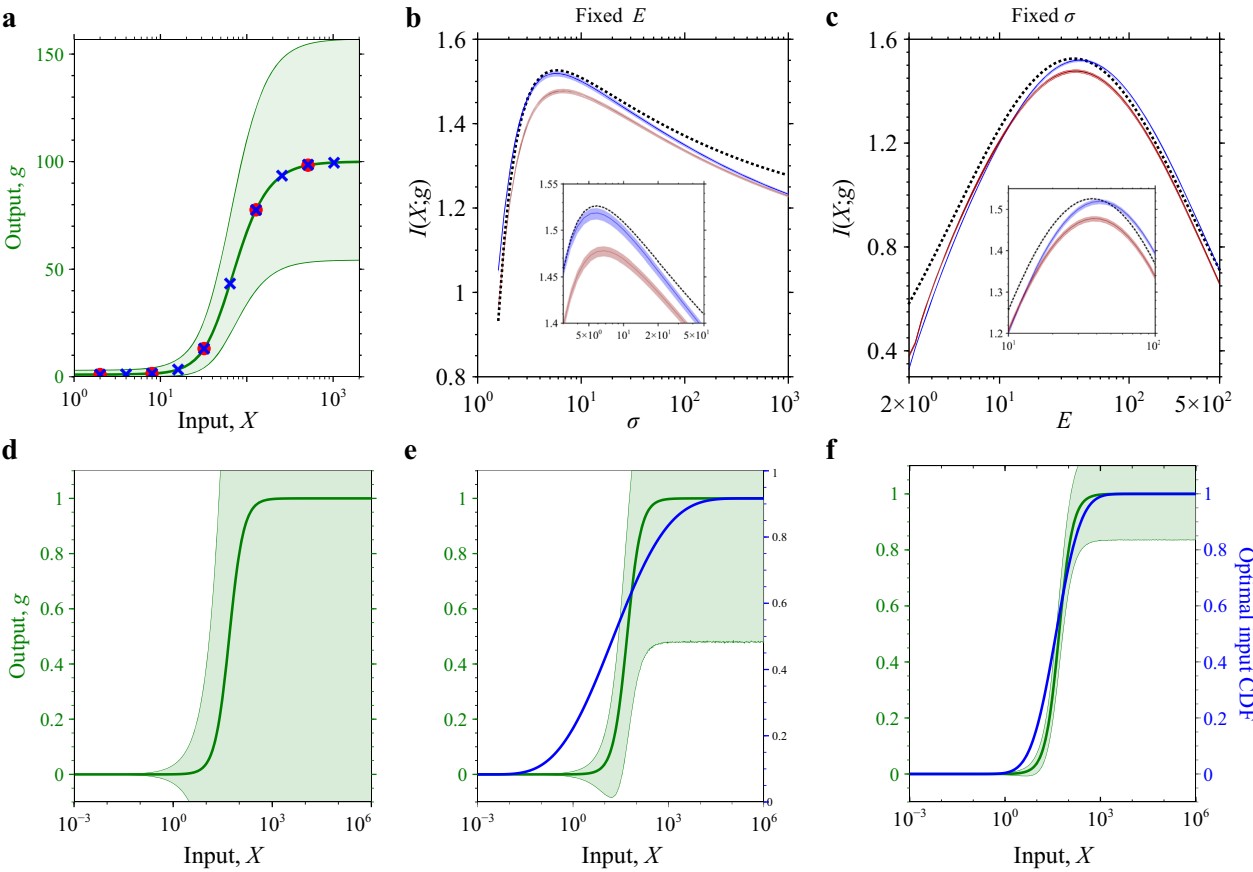

**Fig. 2 Validation of SEMIL using model response functions. a** Model input–output response used for validation of SEMIL. The green line is the mean gene expression input–output response function and the shaded region shows the 5% to 95% quantile range. The point symbols mark the discrete input values used for mutual information estimates with five (red circles) or ten (all markers) input values. **b** Comparison between SEMIL mutual information estimates and the correct mutual information for input distributions with a fixed geometric mean, $E$, and varying geometric standard deviation, $\sigma$. **c** Comparison between SEMIL mutual information estimates and the correct mutual information for input distributions with fixed geometric standard deviation and varying geometric mean. In **b**, **c**, the red curves are the mutual information estimates obtained using SEMIL with mock data at five input values and the blue curves are the estimates obtained with ten input values. The shaded region around each curve bounds the 5% to 95% confidence interval from 100 replicates. The dashed black curve is the correct mutual information calculated using numerical integration. The fixed geometric mean for **b** and the fixed geometric standard deviation for **c** were chosen to match the input distribution at which mutual information is maximum. **d**–**f** Comparison of SEMIL result to the small-noise-limit prediction. The green curves (left $y$-axis) are the mean gene expression output and the shaded region shows the 5% to 95% quantile range. The blue curves (right $y$-axis) show the optimal cumulative distribution function (CDF), corresponding to the input distribution that maximizes the mutual information. The geometric means of the optimal input distributions are 7.76 (**d**), 19.5 (**e**), and 39.8 (**f**). The geometric standard deviations of the optimal input distributions are 100 (**d**), 19 (**e**), and 4.07 (**f**). For comparison, an input distribution with a CDF exactly matched to the mean gene expression output curve would have a geometric mean of 50 and a geometric standard deviation of 2.51.

model response functions, we sampled the output data for the same set of input values. The corresponding optimal input distributions from SEMIL matched the theoretically expected trend of approaching the mean response function with decreasing noise (blue lines in Fig. 2d–f).

**Validation using stochastic simulation data**. To test SEMIL with more realistic output data, we used the Gillespie algorithm to simulate a simple BRN modeled after the lactose sensing system in *Escherichia coli* (see Methods, Supplementary Methods 4, and Supplementary Fig. 2). We simulated output data for a dense set of input values to compute a reference mutual information landscape without SEMIL (Supplementary Fig. 3). Then, we took sparse subsets of the total data, for 5, 10, and 20 discrete input values, and determined the same mutual information landscape using SEMIL. The landscape has a central region of high-mutual information (Fig. 3a–c, panel 2), with well-defined peak that

defines the optimal input distribution (white dots in each plot). The optimal input distribution is approximately matched to the midpoint of the BRN input–output response function, but is generally wider than the response function (Fig. 3a–c, panel 1). In the center of the design space, the accuracy of SEMIL does not depend on the number of input levels used (Fig. 3d–f). Near the boundaries of the design space, the accuracy generally improves with increasing number of input values. This is particularly noticeable along the boundaries corresponding to very narrow or very wide input distributions (top and bottom edges of each plot).

**Mutual information landscapes of engineered BRNs**. To demonstrate the utility of SEMIL for comparing the performance of engineered BRNs, we constructed six different BRNs similar to the lactose sensing system in *E. coli* and used them to systematically examine the effect on the mutual information landscape due to changes in the rate constants and feedback pathways of the

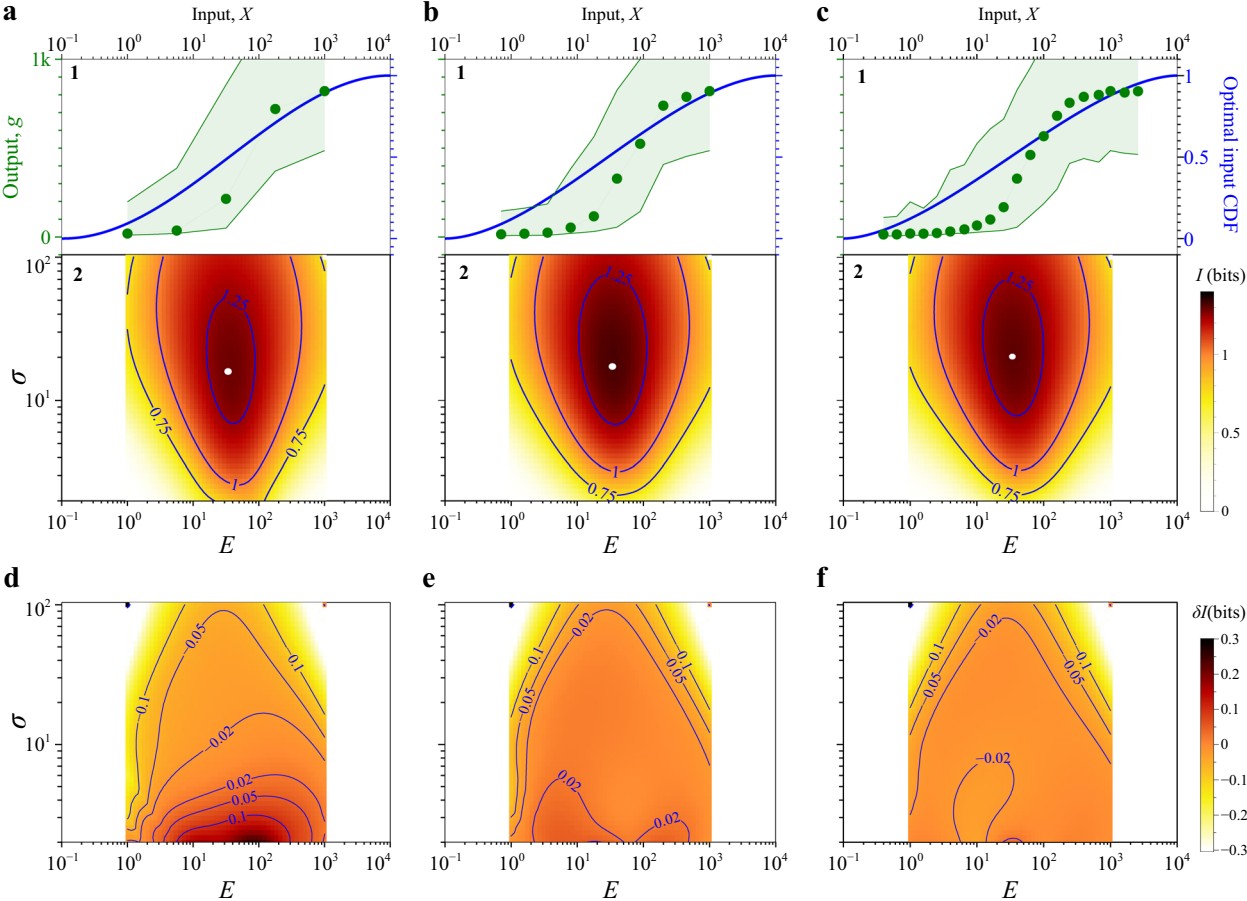

**Fig. 3 Accuracy of mutual information landscapes estimates from SEMIL for simulated BRN data. a–c** Mutual information landscapes from SEMIL obtained using data at 5 (**a**), 10 (**b**), and 20 (**c**) discrete input values. Panel 1 of each subplot shows the mean gene expression input–output response function (green, left $y$-axis) and the cumulative distribution functions (CDFs) of the optimal input distribution (blue, right $y$-axis) for the simulated BRN. The optimal input distribution is the distribution that maximizes the mutual information. The green point symbols indicate the discrete input values used for each case. Panel 2 of each subplot shows the mutual information landscape. Each point in the two-dimensional landscape represents an input distribution with the specified geometric mean, $E$, and geometric standard deviation, $\sigma$. The heat map color indicates the mutual information, $I$, for the input distribution defined by the coordinates ($E$, $\sigma$). The optimal input distribution is marked with a white dot in each landscape plot. The geometric means of the optimal input distributions are the same in all three cases, 33.9. The geometric standard deviations of the optimal input distributions are 15.9 (**a**), 17.3 (**b**), and 20.2 (**c**). **d–f** Accuracy of SEMIL across the design space. The heat map plots show the error in the SEMIL results, $\delta I = I_{SENIL} - I_{correct}$, for mutual information estimates obtained using data at 5 (**d**), 10 (**e**), and 20 (**f**) discrete input values.

BRN (see Fig. 4a and Methods). Specifically, we studied two groups of BRNs: one with the *lacY* gene controlled by the BRN output and one with the *lacY* gene deactivated. When activated, *lacY* increases the transport of the input signal, isopropyl β-D-1-thiogalactopyranoside (IPTG), into the cell, acting as a positive feedback on the BRN output. Within each group, we also varied the rate of translation of the *lacI* repressor by using three different ribosomal binding site (RBS) sequences with predicted translation rates that varied over three orders of magnitude[20]. For each of the six BRNs, we used flow cytometry to measure the distribution of gene expression output at a set of IPTG concentrations. The mutual information landscapes for the engineered BRNs are shown in Fig. 4b–f (replicates in Supplementary Figs. 4–9).

## Discussion

The resulting mutual information landscapes of all six BRNs have some common features (Fig. 4). As with the simulation results, each landscape has a region of high-mutual information that approximately coincides with the midpoint of the BRN response function. In addition, the high-mutual information region is funnel shaped: as the mean of the input distribution varies, the mutual information

changes more steeply for narrow input distributions than for wide input distributions. The mutual information landscapes thus show quantitatively how sensing applications with a narrowly distributed input signal require more precise matching of the BRN response function to the input distribution, because the midpoint of the BRN response function must be near the median of the input distribution to enable high information transmission. Whereas applications with a wide input distribution will require less precise matching since the active range of the BRN response will overlap with the input distribution, providing relatively high information transmission even when the median of the input distribution and midpoint of the BRN response function do not coincide. SEMIL compares favorably with Blahut–Arimoto algorithm (Supplementary Fig. 10), which is the most common existing method to compute the maximum mutual information. However, the corresponding optimal input distribution from Blahut–Arimoto is spiky and discontinuous and difficult to interpret as a biologically plausible input distribution[7]. SEMIL circumvents this problem of interpretation by using a well-defined design space of continuous probability distributions. We also computed the mutual information landscapes for the experimental data using output data for smaller sets of input (Supplementary

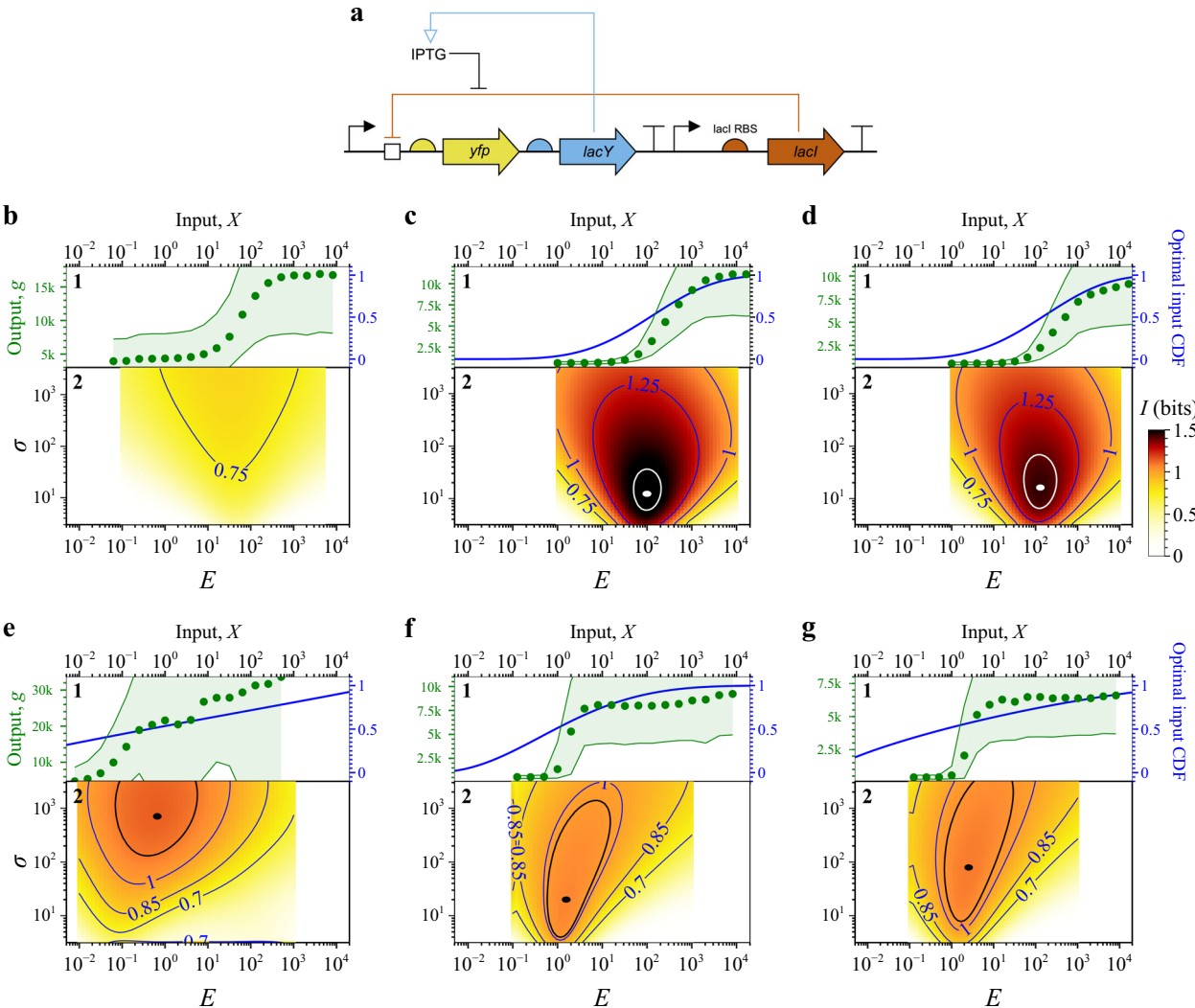

**Fig. 4 Applications of SEMIL to experimentally measured BRNs. a** DNA design schematic[26] of the engineered BRNs used to obtain output data for SEMIL (see Methods). To systematically study the effect of positive feedback on mutual information landscapes, half of the engineered BRNs (data shown in **e–g**) included the *lacY* gene as shown in the diagram, while the other half (data shown in **b–d**) included a deactivated *lacY* gene (Supplementary Methods 5). **b–g** Panel 1 of each plot shows the measured input–output response function (green, left *y*-axis) and the cumulative distribution functions (CDFs) of the optimal input distribution (blue, right *y*-axis) for each BRN. The optimal input distribution is the distribution that maximizes the mutual information. The green point symbols indicate the discrete input values used for each case. Panel 2 of each plot shows the mutual information landscape calculated with SEMIL for each BRN. **b–d** Results for BRNs without the *lacY* feedback with three different translation rates for the *lacI* repressor gene. **e–g** Results for BRNs with the *lacY* feedback for the same three *lacI* translation rates. The estimated relative *lacI* translation rate constants are 0.008 (**b, e**), 1 (**c, f**), and 10 (**d, g**)[20]. For each mutual information landscape, black or white dots indicate the location of the optimal input distribution and the same-colored contours around them bound the mutual information values that are within 0.05 bits of the maximum. Biological replicate data for each BRN is shown in Supplementary Figs. 4–9. For the BRN without *lacY* feedback and the weakest *lacI* translation rate constants (**b**), the maximum mutual information is located at the boundary of the design space for two out of three biological replicates, hence **a** does not show an optimal input distribution.

Figs. 11–16), and found that halving the set of output data did not effect the profile and the magnitude of the landscape.

The shape of the optimal input distribution and the region of high-mutual information both depend on the relative gene expression output noise of the BRN. With small BRN output noise, the mutual information landscape has a clear local maximum at an input distribution with both center and width approximately matched to the BRN response function (Fig. 4c, d). With larger BRN output noise, the optimal input distribution has a higher geometric standard deviation, and the information transmission becomes less sensitive to changes in the width of the input distribution. Positive feedback decreases the width of the response function, but it also increases the BRN output noise. Consequently, we observe that although the BRNs with feedback have a sharper, more step-like

response function, from low to high output, the resulting optimal input distributions are wider than the width of the BRN response function (Fig. 4e–g), or the probability density is less concentrated near the midpoint of the response function. Additionally, BRNs with positive feedback have a flatter mutual information landscape with a larger region of nearly optimal information transmission compared to the BRNs without positive feedback (black or white contours in Fig. 4). In particular, the region of nearly optimal information transmission for some of the BRNs with positive feedback extends across a large range of geometric standard deviation (Fig. 4f–g and Supplementary Figs. 8 and 9).

SEMIL uses ideas from information theory and stochastic modeling to provide a universal quantitative BRN performance metric, which evaluates the information transmission capability of

engineered BRNs across a broad range of possible input environments. Currently, there is considerable interest in understanding the role of information transmission in biology, specifically with regard to metabolic activity, fitness, and phenotypic differentiation. As we demonstrate here, SEMIL can be easily applied to explore these questions using readily available single-cell response data.

## Methods

**Data requirements for SEMIL**. The data needed for SEMIL consists of single-cell gene expression output data measured at different values of an input stimulus. The input stimulus is typically the concentration of a chemical species applied to induce the measured cellular response, but could also be the temperature or any other control parameter that affects the cellular response. One of the key benefits of SEMIL is that it only requires output data for a relatively small number of discrete input stimulus values. The output data can be obtained from simulations or from experiments such as flow cytometry (used in this work), or from other measurements such as single-cell microscopy or single-cell RNA sequencing. At each input stimulus value, accurate mutual information calculations with SEMIL require a sufficient number of single-cell output data points to serve as a model-free estimate of the conditional output probability distributions (see below). For the results shown in Fig. 3, 250,000 points were used for each input value, and for the results in Fig. 4, between 1000 and 50,000 data points were used for each input value.

**Design space for mutual information landscapes**. The first step in analyzing a BRN with SEMIL is to choose the design space, which is the space of input probability distributions, $p(X)$, over which the mutual information will be calculated. The design space can consist of any continuous probability distribution, but if unbounded distributions are used, the errors in the mutual information estimate can be large. Consequently, the design space should be restricted to bounded probability distributions of input values, $X$. For most BRNs, the bounded interval can be chosen based on the application, for example, the minimum and maximum chemical concentrations that might be encountered in a sensing application. If necessary, an unbounded range of input values can be mapped to a bounded interval using, for example, the logistic transform, so the requirement for a bounded interval of input values does not affect the general applicability of SEMIL. In this work, we chose a design space of log-transformed beta distributions on a bounded interval of the input concentration, which enables inclusion of both unimodal and antimodal probability distributions in the same design space. For visualization of the resulting mutual information landscapes, we plot the mutual information values as a heat map vs. the center (geometric mean) and width (geometric standard deviation) of the input distributions. More specifically:

$$X = \text{input concentration},$$
$$p(X) = \frac{1}{X\log(X_{max}/X_{min})}\rho_{beta}\left(\frac{\log_{10}(X/X_{min})}{\log_{10}(X_{max}/X_{min})}\right), \quad (1)$$

where $X_{min}$ and $X_{max}$ are the upper and lower bounds for the input concentration, $X$, and $\rho_{beta}$ is the beta density function for the domain [0, 1].

The input stimulus values used to generate the data for SEMIL need to be matched to the design space so that the range of input values with high probability density is well covered by the set of discrete input values. For example, for the mutual information landscapes shown in Figs. 2 and 3, the input values were chosen to span the range of geometric means used for the design space, and the spacing between different input values was chosen to match the minimum geometric standard deviation used for the design space.

**Stochastic reduced-order model of input distribution**. The second step of SEMIL is to find the best discrete approximation for each continuous input distribution in the design space. For each input distribution in the design space, stochastic reduced-order modeling is used to find the optimal probability masses to assign to the available set of discrete input values so that the resulting discrete probability distribution best represents the continuous input distribution. The best discrete approximation is found by minimizing the sum of two error functions, one given by the integral of the squared error of the cumulative distribution and the second given by the sum of squared errors of the distribution moments (Supplementary Methods 1).

**Simulations**. Simulated BRN output data used for validation of SEMIL (Fig. 3a–f) were obtained using the Gillespie algorithm to simulate the output response from a reaction network model of the *E. coli* lactose (*lac*) operon[21] (Supplementary Methods 4 and Supplementary Tables 1–4).

**Calculation of mutual information**. SEMIL computes the mutual information of a BRN for each input distribution in the design space using discrete approximations of the probability distributions of the input, $X$, and the output, $g$. The stochastic reduced-order model maps each continuous input distribution $p(X)$, to a discrete

distribution of the input, $P(X = x_i)$, where the input, $X$, takes a set of fixed values $\{x_i\}$. The BRN output data for each of the fixed input values, $X = x_i$, is used to estimate discrete conditional probabilities of the output, $P(g = g_j|X = x_i)$ for the set of possible output values, $\{g_j\}$. For naturally discrete output data (e.g., molecular counts from the Gillespie simulations), the set of possible output values is taken to be the set of non-negative integers. For continuous output data (e.g., flow cytometry data), the output values are binned following a procedure to minimize bias in the mutual information estimates (Supplementary Methods 6). To avoid the need for additional assumptions, the observed frequencies are used directly as the estimates for the discrete conditional probabilities used to calculate the mutual information, $I$:

$$I(X;g) = \sum_i P(X = x_i)H(g|X = x_i) + H(g), \quad (2)$$

where $H(g)$ is the entropy of the associated marginal output distribution

$$H(g) = -\sum_j P(g = g_j)\log_2 P(g = g_j), \quad (3a)$$

$$P(g = g_j) = \sum_i P(X = x_i)P(g = g_j|X = x_i), \quad (3b)$$

and where $H(g|X = x_i)$ is the entropy associated with the conditional output distribution,

$$H(g|X = x_i) = -\sum_j P(g = g_j|X = x_i)\log_2 P(g = g_j|X = x_i). \quad (4)$$

Estimates of mutual information directly from finite data as described here can be systematically biased. To correct for this bias, for every point in the design space, we extrapolate to the limit of infinite data using previously described methods[4,7,16] (Supplementary Methods 6 and Supplementary Fig. 17).

**Strain construction**. Wild-type *E. coli* strain MG1655 was purchased from the American Type Culture Collection (ATCC 47076). The strain MG1655Δ*lac* was constructed by replacing the genomic copy of the *lac* operon, comprising genes *lacIZYA*, with the bleomycin resistance protein from *Streptoalloteichus hindustanus* (*Shble*). The *Shble* cassette was synthesized by Integrated DNA Technologies as a gBlock, and was codon optimized for expression in *E. coli* and placed under control of the constitutive promoter J23101 and the RiboJ ribozyme insulator[22]. The *Shble* cassette was inserted into the genome of *E. coli* MG1655 using recombineering to facilitate homologous recombiation as described below.

*Escherichia coli* MG1655 was transformed with the recombineering plasmid pSIM29, described elsewhere[23]. Briefly, pSIM29 contains λ phage genes *Exo*, *Beta*, and *Gam* under the control of a temperature-dependent repressor (cI857) and temperature-sensitive origin of replication. Plasmid pSIM29 was maintained in MG1655 using lysogeny broth (LB) supplemented with hygromycin (200 μg mL$^{-1}$) and grown at 30 °C.

The *Shble* cassette was amplified by PCR using primers DT.01 and DT.02 (Supplementary Methods 5), which included 50 bp of homology to the genome sequence flanking the *lac* operon of *E. coli* MG1655. Overnight culture of *E. coli* MG1655 containing pSIM29 was diluted 1000-fold and grown at 30 °C in LB to mid-log phase (5 h). Culture was then submerged in 44 °C water bath for 1 h, followed immediately by a 30-min incubation in ice slurry. Cultures were pelleted and washed with 10% glycerol twice, and electroporated to transform the amplified *Shble* cassette. Cultures were recovered with Super Optimal broth with catabolite repression at 37 °C for 3 h, and then plated on LB-agar supplemented with zeocin (50 μg mL$^{-1}$).

The following day, colonies were screened using colony PCR to verify insertion of the *Shble* cassette. Two colonies were genome sequenced using the Illumina MiSeq platform, with 600 cycle chemistry (2 × 300, paired ends). Reads were aligned to the MG1655 published genome NC000913.3 using breseq program[24]. Sequencing results verified successful substitution of the *lac* operon with the *Shble* cassette.

**Plasmid construction**. Plasmids used for the results shown in Fig. 4 were constructed from the plasmid pAN1818[1]. Plasmid pAN1818 encodes YFP under the control of the *tacI* promoter and regulated by the *symL* (*lacO*) operator; pAN1818 also contains the kanamycin resistance marker, and p15A origin of replication (Supplementary Fig. 18).

To evaluate feedback effects, the *lacY* gene was added to pAN1818 downstream of the YFP cassette. The *lacY* gene was amplified by PCR from the genome of *E. coli* MG1655 using primers DT.03 and DT.04. Plasmid pAN1818 was amplified by PCR using primers DT.05 and DT.06. The plasmid amplicon and *lacY* amplicon were combined with Gibson assembly and sequence verified with Sanger sequencing. The resulting plasmid was used for the results shown in Fig. 4.

The plasmid with increased *lacI* translation rate (Fig. 4d, g) was constructed by changing the *lacI* RBS (Supplementary Table 5). The plasmid was amplified by PCR using primers DT.07 and DT.08. The resulting amplicon was recircularized

with Gibson assembly. Plasmids containing the correct sequence were verified with Sanger sequencing.

The plasmid with decreased *lacI* translation rate (Fig. 4b, e) was constructed by changing the *lacI* RBS. The plasmid was amplified by PCR using primers DT.09 and DT.10. The resulting amplicon was circularized with Gibson assembly in the presence of primer DT.11. Plasmids containing the correct sequence were verified with Sanger sequencing.

Plasmid constructs without feedback (Fig. 4b–d) were generated by inserting three sequential stop codons at positions D35 I36 N37 into the *lacY* gene encoded on the plasmid. The plasmids were amplified by PCR using primers DT.12 and DT.13. The resulting amplicon was circularized with Gibson assembly. Plasmids containing the correct sequence were verified with Sanger sequencing. All plasmid constructs were electroporated into MG1655 for routine cloning. Plasmids were sequence verified using Sanger sequencing (Psomagen USA).

**Flow cytometry**. Two or three biological replicates of MG1655Δ*lac* with each plasmid were grown overnight to stationary phase in M9-glucose media (Supplementary Tables 6 and 7). Each replicate was then diluted 1000-fold into a 96-well assay plate (4titude, 4ti-0255) containing 500 μL M9-glucose media per well with various concentrations of IPTG, ranging from 0 to 2.048 mmol L$^{-1}$. For each biological replicate, three technical replicates were included in each plate. Plates were sealed with clear gas-permeable seals (4titude, 4ti-0541/SLIT) and cultures were grown for 3.5 h with double orbital shaking in BioTek Epoch 2 plate readers at 37 °C. Optical density at 600 nm (OD600) was monitored during growth, and typical OD600 values at the end of 3.5 h were between 0.04 and 0.07. After growth, 20 μL samples from each well were diluted into 180 μL of phosphate-buffered saline supplemented with 170 μg mL$^{-1}$ chloramphenicol to halt protein translation. The resulting diluted samples were measured on an Attune NxT flow cytometer with 488 nm excitation laser and a 530 ± 15 nm bandpass emission filter. Blank samples were measured with each set of *E. coli* samples, and the results of the blank measurements were used with an automated gating algorithm to discriminate cell events from non-cell events. A second automated gating algorithm was used to select singlet cell events and exclude doublet, triplet, and higher-order multiplet cell events. All subsequent analysis was performed using the singlet cell event data (Supplementary Fig. 19).

**Statistics and reproducibility**. For each experimental BRN, we measured either two or three biological replicates (labeled A, B, and C), and for each biological replicate, we measured three technical replicates (labeled 1, 2, and 3). We computed the mutual information landscapes for each of the replicates, which are shown in Supplementary Figs. 5–10. Estimated mutual information was corrected for finite-sampling bias by systematic subsampling of the experimental data and extrapolating to unbiased mutual information values using existing methods (Supplementary Methods 6)[7].

**Reporting summary**. Further information on research design is available in the Nature Research Reporting Summary linked to this article.

## Data availability

The parameters for the numerical studies and Gillespie simulations are present in the Supplementary Tables 1–4. The data for the main figures are in Supplementary Data 1, 2, and 3. The flow cytometry data and all other data are available upon request. The plasmid backbone used for this work has been deposited in GenBank, with the accession code MT012365. Physical plasmids and their sequences, as well as *E. coli* strain MG1655Δ*lac*, have been deposited to Addgene.

## Code availability

The code to compute mutual information landscapes is available at https://github.com/usnistgov/InGene[25].

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

## Acknowledgements

We acknowledge financial support from the NIST Innovations in Measurement Science Program. This work was supported by National Research Council Post-Doctoral Fellowships to D.T. (Certain commercial equipment, instruments, or materials are identified in this paper in order to specify the experimental procedure adequately. Such identification is not intended to imply recommendation or endorsement by the National Institute of Standards and Technology, nor is it intended to imply that the materials or equipment identified are necessarily the best available for the purpose.)

## Author contributions

S.S. developed SEMIL, performed stochastic simulations, and computations for all the figures. D.T. engineered the synthetic BRN and performed flow cytometry measurements. D.R. performed the gating analysis of the flow cytometry data. All three authors contributed to the writing of the manuscript.

## Competing interests

The authors declare no competing interests.
