## [Peer Review File · Communications Biology]

Reviewers' comments:

Reviewer #1 (Remarks to the Author):

Reviewer #2 (Remarks to the Author):

In this manuscript, the authors proposed a method of sparse estimation of mutual information landscapes to quantify information transmission through cellular BRNs. The method is very interesting and looks solid.

Some major comments are below:

1. The authors need to give more discussions on the existing methods to compute mutual information compared with the proposed method, and explicitly discuss the significance of the work. For example, any comparison of six experimentally-measured BRNs with other works?
2. For experimentally-measured BRNs, the mutual information landscapes are plotted for many biological replicates in supplementary information. Are they almost the same if the authors use more (or less) input values, for example 5 or 20 input values, to calculate the mutual information landscape?
3. The reason of choosing the beta distribution as the continuous input distribution is not mentioned in this paper.

We appreciate the reviewer's feedback and we have included new figures and text to address the comments. Here are our specific responses to the major comments raised by the reviewer.

Reviewer 2:

Some major comments are below:

1. The authors need to give more discussions on the existing methods to compute mutual information compared with the proposed method, and explicitly discuss the significance of the work. For example, any comparison of six experimentally-measured BRNs with other works?

Response: We agree that comparison with other methods would clarify the impact of our work described in this manuscript. We have included new figures comparing SEMIL with existing methods and results:

- a) The most common algorithm for computing the maximum mutual information is the Blahut-Arimoto algorithm. We have compared the maximum mutual information from SEMIL with the result from Blahut-Arimoto algorithm (Supplementary Information: Section 6, subsection Maximum Mutual information from SEMIL and Blahut-Arimoto algorithm).

The Blahut-Arimoto algorithm performs an unconstrained search, whereas SEMIL constrains the search across the chosen *design space* of continuous input distributions. Hence, the value from SEMIL is close to but consistently lower compared to the Blahut-Arimoto algorithm. But the optimal input distribution from the Blahut-Arimoto algorithm is spiky and discontinuous and difficult interpret as a biologically relevant distribution [Ref. 7], whereas SEMIL operates on a space of continuous probability distributions. We have included new text (lines 108-112 in the manuscript) stating this important advantage of SEMIL.

- b) We have shown in Fig. 2d-2f of the manuscript that the optimal input distribution computed using SEMIL approaches the theoretically-predicted solution in the small noise limit (lines 62-70 in the manuscript).

2. For experimentally-measured BRNs, the mutual information landscapes are plotted for many biological replicates in supplementary information. Are they almost the same if the authors use more (or less) input values, for example 5 or 20 input values, to calculate the mutual information landscape?

Response: We have included new figures (Supplementary Information: Section 6, subsection Mutual information landscapes using smaller sets of input values) comparing the mutual information landscapes obtained using one-fourth (5) and half (9) of the input values to the landscape from the complete set (18) of input values. For all the six types of experimentally-measured BRNs the mutual information landscapes obtained using the half and the complete set of input values are the same.

3. The reason of choosing the beta distribution as the continuous input distribution is not mentioned in this paper.

Response: The need for choosing bounded distributions is already described in Methods. We chose beta distribution because it can be both a unimodal and an antimodal distribution in different regions of the same *design space*. We have added this reasoning in Methods under the subsection Design space for mutual information landscapes (lines 192-194).

REVIEWERS' COMMENTS:

Reviewer #2 (Remarks to the Author):

The authors have addressed all of my previous comments. I support the publication of this manuscript.